# Impact of bread diet on intestinal dysbiosis and irritable bowel syndrome symptoms in quiescent ulcerative colitis: A pilot study

Aleix Lluansí[1]*, Marc Llirós[1¤], Robert Carreras-Torres[1], Anna Bahí[1], Montserrat Capdevila[1], Anna Feliu[1], Laura Vilà-Quintana[1], Núria Elias-Masiques[2], Emilio Cueva[2], Laia Peries[1,3], Leyanira Torrealba[1,3], Josep Oriol Miquel-Cusachs[1,3], Míriam Sàbat[1,4], David Busquets[1,3], Carmen López[1,3], Sílvia Delgado-Aros[5], Librado Jesús Garcia-Gil[1,6], Isidre Elias[3], Xavier Aldeguer[1]*

1 Digestive Diseases and Microbiota Group, Institut d'Investigació Biomèdica de Girona Dr. Josep Trueta (IDIBGI), Girona, Spain, 2 Elias-Boulanger S.L., Vilassar de Mar, Spain, 3 Department of Gastroenterology, Hospital Universitari de Girona Dr. Josep Trueta, Girona, Spain, 4 Department of Gastroenterology, Hospital de Santa Caterina, Girona, Spain, 5 Gastroenterology Scientific advisor to Elias-Boulanger S.L., Vilassar de Mar, Spain, 6 Department of Biology, Universitat de Girona, Girona, Spain

¤ Current address: Bioinformatics and Bioimaging (BI-SQUARED) research group, Biosciences Department, Faculty of Sciences, Technology and Engineerings, Universitat de Vic–Universitat Central de Catalunya, Vic, Catalunya, Spain

* xaldeguer@idibgi.org (XA); alluansi@idibgi.org (AL)

**Data Availability Statement:** All sequencing data files are available from the DDBJ/EMBL/GenBank database at accession number PRJNA902141.

## Abstract

Gut microbiota may be involved in the presence of irritable bowel syndrome (IBS)-like symptomatology in ulcerative colitis (UC) patients in remission. Bread is an important source of dietary fiber, and a potential prebiotic. To assess the effect of a bread baked using traditional elaboration, in comparison with using modern elaboration procedures, in changing the gut microbiota and relieving IBS-like symptoms in patients with quiescent ulcerative colitis. Thirty-one UC patients in remission with IBS-like symptoms were randomly assigned to a dietary intervention with 200 g/d of either treatment or control bread for 8 weeks. Clinical symptomatology was tested using questionnaires and inflammatory parameters. Changes in fecal microbiota composition were assessed by high-throughput sequencing of the 16S rRNA gene. A decrease in IBS-like symptomatology was observed after both the treatment and control bread interventions as reductions in IBS-Symptom Severity Score values (p-value < 0.001) and presence of abdominal pain (p-value < 0.001). The treatment bread suggestively reduced the Firmicutes/Bacteroidetes ratio (p-value = 0.058). In addition, the Firmicutes/Bacteroidetes ratio seemed to be associated with improving IBS-like symptoms as suggested by a slight decrease in patient without abdominal pain (p-value = 0.059). No statistically significant differential abundances were found at any taxonomic level. The intake of a bread baked using traditional elaboration decreased the Firmicutes/Bacteroidetes ratio, which seemed to be associated with improving IBS-like symptoms in quiescent ulcerative colitis patients. These findings suggest that the traditional bread elaboration has a potential prebiotic effect improving gut health (ClinicalTrials.gov ID number of study: NCT05656391).

**Funding:** This study was funded by the Ministry of Economy, Industry and Competitiveness (MINECO) RETOS program (RTC-2017-6467-2). AL benefits from a grant included within the RTC-2017 program. The Instituto de Salud Carlos III supported RCT through the Miguel Servet Program CP21/00058. IE, NE, SDA and EC are employees of Elias-Boulanger, who have received funding from RTC-2017 program. The funders had no role in study design, data collection and analysis, decision to publish or manuscript preparation.

**Competing interests:** The authors have declared that no competing interests exist.

## Introduction

Ulcerative colitis (UC) is a chronic idiopathic inflammatory bowel disorder of the colon that causes continuous mucosal inflammation extending from the rectum to the more proximal colon, with variable extents. UC evolves throughout episodes of relapses or flare-ups, when symptoms become more active, and periods of remission [1].

Despite the absence of inflammatory disease activity, some UC patients in remission report gastrointestinal (GI) symptoms compatible with Irritable Bowel Syndrome (IBS), often referred to as IBS-like symptoms [2]. According to the updated Rome IV criteria [3, 4], IBS has four subtypes: (i) IBS with constipation (IBS-C); (ii) IBS with diarrhea (IBS-D); (iii) mixed IBS (IBS-M); and (iv) unclassified IBS. Abdominal pain, straining, myalgia, urgency, bloating, and feelings of serious illness are reported by patients as the most distressing symptoms [5]. In addition, psychological factors (*e.g.*, increased anxiety, depression, and reduced quality of life, among others) have been associated with IBS-like symptoms [2, 6–8]. The absolute global prevalence of IBS, based on Rome IV criteria, was estimated at 5.0–6.0% [9]. However, the pooled prevalence of IBS-like symptoms in inflammatory bowel disease (IBD) patients in remission was 32.5%, as observed in a recent systematic review and meta-analysis that included 27 studies and 3169 patients [10], highlighting the importance of a such group of patients.

The causes of IBS-like symptoms in UC patients are still unknown [2, 6, 7, 11, 12]. However, some reports suggested that persisting occult low-grade intestinal inflammatory disease activity may be involved [6, 7, 12, 13]. Therefore, gut-impaired homeostasis or dysbiosis, including increased intestinal permeability, altered immunologic pathways, and microbiota profiles, may be behind this symptomatology. The role of gut microbiota in the onset and per-petuation of intestinal inflammation in IBD [14, 15] and IBS [16, 17] has been studied over the last decade. Numerous studies have revealed that the composition of the fecal microbiota of IBD patients, IBS patients, and healthy controls differs [18–21] with a significant decrease in microbial richness linked to increased disease severity [22–25]. In recent years, a potential gut dysbiosis marker (*i.e.*, the ratio between Firmicutes and Bacteroidetes phyla; F/B ratio) has been proposed and found to be increased in IBS [26–29] or obesity patients [30], and decreased in IBD patients [30, 31] compared to healthy controls. However, novel microbiota sequencing techniques, such as high-throughput sequencing of the 16S rRNA gene or whole genome sequencing, encourage the search for more robust markers, thus, identifying potential underlying factors which might allow the development of tailored therapies to restore gut homeostasis.

Among the strategies to modulate the intestinal microbiota composition, prebiotic products include the consumption of non-digestible food for humans, such as fiber, which is degraded by some commensal bacteria within the colon. Prebiotic products selectively stimulate bacterial metabolic pathways that produce key metabolites, such as anti-inflammatory short-chain fatty acids [32, 33]. In particular, bread is a relevant component of the Western diet and an important source of dietary fiber [34]. The bread making process, such as dough composition and fermentation, impacts bread's final chemical composition [35–37]. In addition, it has been observed that sourdough, long-fermentation and whole-grain flour might modulate gut microbiota and lower systematic inflammation [34, 36, 38, 39]. Finally, in a previous work of our group, *in vitro* experiments characterized the potential of different types of bread to modulate stool microbial composition in IBD patients towards a microbiota profile closer to healthy subjects [37].

The present study aimed to compare the *in vivo* prebiotic properties of bread produced by traditional bread-making techniques with that made using a modern bread making method on

IBS-like symptoms of patients with quiescent UC. The expected outcome of the differential effects was a change in the fecal microbiome composition, which may indicate changes in the mucosa-associated microbiota.

## Materials and methods

### Study design

The present study was designed as a randomized, double-blind, parallel-group pilot clinical trial on UC patients suffering from IBS-like symptoms to determine the impact of a traditional bread-based dietary intervention to modulate intestinal dysbiosis and relieve symptoms (S1 Checklist). Up to twenty-three subjects were included in the study. Control subjects were not included in the present study, as the primary focus was to investigate differences in patients' symptomatology. The dietary intervention consisted of a daily consumption of 200 grams of either treatment or control bread for eight weeks. No other changes in subjects' usual medication, diet, and lifestyle were requested for the study. Furthermore, subjects were asked not to alter their diet during the intervention period. Participants were randomly assigned to the treatment or the control group in a 1:1 ratio based on a computer-generated randomization schedule. An independent, non-involved person in the study generated the randomization list, so randomization was blinded for both the participants and the investigators.

A baseline visit prior to intervention start (*i.e.*, week 0) and a visit at the end of the study (EOS) (*i.e.*, week 8) were performed. At both visits, patients were requested to deliver two fresh fecal samples in sterile containers from the same deposition, previously collected at home, within four hours before the visit. One fecal sample was used to analyze fecal calprotectin concentration at HUJT local laboratory as per clinical practice, and the other sample was stored at -80˚C until further processing for microbial composition analyses. In addition, blood samples for biochemistry tests (*i.e.*, albumin, triglycerides, cholesterol, protein C-reactive, creatinine, sodium, potassium, and calcium) and hematology tests (*i.e.*, hemogram and erythrocyte sedimentation rate (ESR)) were collected and analyzed at HUJT local laboratory as per common clinical practice. S1 Fig depicts the study design.

At baseline and EOS visits, questionnaires were performed to assess clinical remission of UC and IBS-like symptomatology. These included the partial mayo score [40], Rome IV criteria [41] and IBS-Symptom Severity Score (IBS-SSS) [42] questionnaires. The overall IBS-SSS score was calculated by totaling the punctuation of its five items. Each ranged from 0 to 100: (i) abdominal pain, (ii) number of days of abdominal pain during the last 10 days (number of days with abdominal pain x10), (ii) abdominal distension, (iv) satisfaction of defecatory behavior, (v) interference of IBS symptoms in life. The possible range was 0–500 points.

In addition, the Hospital Anxiety and Depression Scale (HADS) as described by Johnston and co-workers [43] and the 14-item Mediterranean diet adherence as described by Martínez-González and collaborators [44] were performed at baseline visit. These questionnaires were used to appraise symptoms of anxiety and depression and the adherence of participants to the Mediterranean diet. HADS questionnaire evidenced levels of anxiety and/or depression in responders throughout a collection of seven questions for each characteristic, representing independent psychopathological symptoms. Each item was valued on a four-point frequency scale ranging from 0 to 3. On the other hand, the questionnaire evaluated adherence to the Mediterranean diet through 14 questions, accounting for 1 point each. A score of less than nine points was considered low adherence to the Mediterranean diet, while a score equal to or greater than nine was considered high adherence.

**Table 1. Nutritional composition of treatment and control bread types.**

| Components | Treatment bread (n = 3) | Control bread (n = 3) | p-value |
|---|---|---|---|
| Proteins (g/100 g sample) | 8.98 (8.49–9.64) | 10.10 (10.03–10.20) | 0.262 |
| Total fats (g/100 g sample) | 1.11 ± 0.18 | 0.75 ± 0.07 | 0.064 |
| Dietary fiber (g/100 g sample) | 3.27 ± 0.47 | 2.60 ± 0.17 | 0.122 |
| Moisture (g/100 g sample) | 36.00 ± 0.78 | 33.60 ± 0.50 | 0.769 |
| Ash (g/100 g sample) | 2.32 ± 0.61 | 1.59 ± 0.03 | 0.022* |
| Carbohydrates (g/100 g sample) | 49.43 ± 1.17 | 51.47 ± 0.06 | 0.658 |
| Calories (µg/L sample) | 245.33 5.51 | 257.67 ± 1.53 | 0.364 |
| Sodium (g/100 g sample) | 0.63 (0.61–0.67) | 0.50 (0.50–0.51) | 0.042* |
| Potassium (g/kg sample) | 1.57 ± 0.06 | 1.27 ± 0.06 | 0.017* |
| Magnesium (g/Kg sample) | 0.40 ± 0.01 | 0.29 ± 0.01 | 0.036* |
| Calcium (g/Kg sample) | 0.27 ± 0.00 | 0.20 ± 0.00 | 0.039* |

Values are means ± standard deviation (SD) for normally distributed data and medians (inter-quartil range; IQR) for non-normally distributed data. Statistical significant differences were tested using $t$-test for data with normal distribution and Wilcoxon rank-sum test for data with non-normal distribution. * $p$-value $\leq 0.05$.

Furthermore, the subjective improvement or worsening of gastrointestinal symptoms was evaluated at EOS visit through the Likert scale [45], which ranged from 1 to 5 (1, much worse; 2, worse; 3, equal; 4, improvement; and 5, much improvement) (S1 Protocol).

## Subjects

A total of 31 patients with UC in clinical remission who were experiencing IBS-like symptomatology were recruited from the Department of Gastroenterology at the Hospital Universitari Dr. Josep Trueta (HUJT; Girona, Spain) and the Gastroenterology Unit at the Hospital Santa Caterina (HSC; Salt, Girona, Spain). Subjects participated in the study from December 2019 to August 2021. Patients were diagnosed with UC according to established clinical and histological criteria as common clinical practice. Remission was defined as a total Mayo score $\leq 2$ and fecal calprotectin values under 250 µg/g. The inclusion criteria to participate in the study were subjects aged over 18 years who had moderate-to-severe IBS-like symptomatology defined by Rome IV criteria and IBS Symptom Severity Score (IBS-SSS) > 175. Exclusion criteria included the presence of flare-up of UC, coeliac disease, colectomy, or intestinal resection; antibiotic intake, prebiotic or probiotic treatment within 3 months before the study, any malignancy, pregnancy, or breastfeeding, intake of medication potentially influencing gastrointestinal function; and disability to give informed consent.

Participants were asked to answer a questionnaire to record clinical and epidemiological data at recruitment and at the end of the 8 weeks trial. All inclusion and exclusion criteria were assessed at screening visits and written informed consent was given to gastroenterologist investigators. All enrolled participants were asked to not change their dietary behaviour during the study except for the bread intervention. Ethics Committee approval was received for the study, which has been registered with ClinicalTrials.gov (NCT05656391).

## Bread composition and processing

In the present study, we compared the effect of a bread product following a traditional bread-making technique (Elias-Boulanger (EB) treatment bread) in comparison to a modern bread making method (EB control bread) as prebiotic complements to induce benefits in UC patients. The treatment bread was produced following a traditional process consisting of a

sourdough bread prepared with a sourdough starter (30%) generated following baker's receipt (*i.e.*, starter inoculum (T110 flour, *Triticum dicoccoides*; Moulin de Colagne®, France; 70%) and water (30%)), water (50%), and whole-grain wheat flour (T110 flour, *T. dicoccoides;* Moulin de Colagne®, France; 20%). The initial starter was back-slopped for seven days and afterwards mixed (wt/wt; 30% flour basis) with whole-grain wheat flour (*Triticum aestivum*) pressed with the stone mill "Moulin de Coulagne," tap water (wt/wt; 40% flour basis), dry baking yeast (*Saccharomyces cerevisiae*, wt/wt; <0.5% flour basis; Lesaffre (Hirondelle®)), and salt (wt/wt; 1.1% flour basis; Guerande®). Afterwards, a double-stage fermentation process was allowed for 72 hours, and fermented doughs were baked at 200˚C for 90 min in a refractory sole oven (Eurofours®). More details on bread elaboration can be found in a previous work [37], where study bread appears as EB long-fermentation bread 1 (eblfb1).

Control bread was crafted following modern methods and incorporating the typical chemical composition found in commercial bread commonly available in the region. Briefly, this bread was prepared by mixing refined wheat flour (Farinera Corominas®), sourdough starter (wt/wt; 10% flour basis), tap water (wt/wt; 40% flour basis), salt (wt/wt; 1.2% flour basis, Sal Costa®), yeast (*S. cerevisiae*, 0.010 g/kg of flour; Lesaffre (Hirondelle®)), xanthan gum (0.005 g/kg of flour), wheat gluten (0.010 g/kg of flour; Uniplus®), enzymes (α-amylase, endoxylanase, amyloglucosidanase; Uniplus®), and additive components (emulsifier E471, antioxidant E-300; Uniplus®). This bread was fermented for 2 hours and baked at 200˚C for 90 minutes in a refractory sole oven (Eurofours®)).The nutritional composition of both types of bread is listed in Table 1.

Elias Boulanger SL (Vilassar de Mar, Spain) developed both types of bread with similar appearance and taste and supplied them to participants in transparent plastic pouches to reduce aesthetic bias.

## Statistical analyses of blood tests and questionnaire data

Sample size calculations were made using data derived from previous research on other dietary interventions [46], which used the difference in IBS-SSS as the primary end point (a mean of the differences of 50 in IBS-SSS between the pairs). Assuming the standard deviation (SD) of the differences to be 70 and a power of 80% (*t*-test) and an α-error of 5%, a sample size of 18 participants would be needed for each arm. When interpreting the results of the present study it is important to note that 23 out of 31 enrolled subjects completed the trial. Accordingly, our study should be considered as a pilot study that can serve as a foundational exploration into the potential use of bread-based diets for treating IBS symptoms in UC in the future.

Normality for numerical data was assessed through the Shapiro-Wilk test. For data with normal distribution, a *t*-test was used to determine whether there was a statistically significant difference, whereas, for data with non-normal distribution, the Wilcoxon rank-sum test was used. Pearson's chi-squared test was used to determine whether there was a statistically significant difference in categorical data. To test for statistical differences in clinical variables and demographic data after intervention, we used Linear Mixed Models (LMM) and Generalized Linear Mixed Models (GLMM) for numerical and categorical data, respectively. In our statistical approach, we incorporated a random intercept, assuming a variance components structure, accounting for subject-specific variability, and included the interaction Intervention x Time to assess potential interaction effects. Subsequently, in cases where the interaction term was found to be non-significant, we refined the model by excluding the interaction term, to assess main effects. Our statistical analyses primarily relied on *p*-values as indicators of statistical significance when *p*-value ≤ 0.05. All statistical analyses were performed using R software (2.14.0, http://www.r-project.org/).

## Microbiome 16S rRNA gene sequencing and quality control

Total genomic DNA was purified from 200–300 mg fecal samples collected at baseline and EOS visits. DNA extraction was performed using the NucleoSpin Soil Kit (Macherey-Nagel GMbH& Co., Duren, Germany) following manufacturer instructions and eluted in 100 µL of Elution Buffer. Total genomic DNA was quantified using a Nanodrop ND-2000 UV-Vis spectrophotometer (Nanodrop, DE) and a Qubit® (ThermoFisher Scientific®) fluorimeter.

The V3-V4 region of the bacterial 16S rRNA gene was amplified from each sample following standard practices at external facilities (StarSEQ GmbH, www.starseq.com) by October 2021. Briefly, triplicate end-point PCR reactions for V3–V4 region were performed using previously described primers (V3–V4 16S rRNA gene region, 341f–806r [47]) and equimolarly pooled to reduce bias, and finally spiked with 8.5–10% PhiX before sequencing. Paired-End (2 x 300 base pair) high-throughput DNA sequencing was carried out using the MiSeq platform (Illumina®). Obtained sequencing reads were processed with the following packages: DADA2 pipeline and phyloseq R [48, 49]. Default settings were used for filtering and trimming low-quality tails and removing chimaeras. Built-in training models were used to learn error rates for the amplicon dataset. Identical sequencing reads were combined through DADA2's dereplication functionality, and the DADA2 sequence–variant inference algorithm was applied to each dataset. Subsequently, paired-end reads were merged. After DADA2 denoising, removal of chimaeras and filtering, the 46 samples included in the sequencing analysis provided a mean of 98,087.72 reads with an Inter-Quartile Range of 57,683–165,379. The DADA2 pipeline resulted in 918 features (amplicon sequence variants, ASVs). Finally, taxonomy at the species level was assigned to ASVs with a 100% identity using the Silva taxonomic training data version 138 (http://www.arb-silva.de/) and only ASVs assigned to a phylum of the Bacteria domain were kept. We also checked for outliers through a Non-metric Multi-Dimensional scaling (NMDS) using Bray-Curtis distances, and no outliers at six standard deviations of the first and second components were found. Raw reads data and associated metadata have been submitted to the DDBJ/EMBL/GenBank databases under accession number PRJNA902141.

## Statistical analyses of microbiome sequencing data

Alpha diversity indices together with beta-diversity matrices were computed using DADA2 [49], phyloseq [48], vegan [50], ape [51], and phangorn [52] R packages. Also, tidyverse [53], readxl [54], devtools [55] and biostrings [56] libraries of the R software package were used throughout the pipeline. For alpha diversity, Chao1 and Shannon indices were computed. Normality was assessed through the Shapiro-Wilk test, and statistical differences in each treatment group were analyzed using Paired t-test for the Chao1 index and Wilcoxon rank sum exact test for the Shannon index. For beta diversity matrices, unweighted Unifrac, weighted Unifrac and Bray-Curtis distances were computed and plotted through principal coordinate analyses (PCoA). In addition, Aitchison distances were calculated to perform a diversity analysis appropriate to the compositional nature of sequencing data [57, 58] and plotted through principal component analyses (PCA). A non-parametric PERMANOVA test [59], implemented in the adonis function of the R/vegan package (v2.6–2), using 10,000 permutations was then performed to identify statistical distance differences between groups. In addition, to identify whether collected parameters (*e.g.*, demographics, clinical data and diet) were associated with diversity indexes, a canonical correspondence analysis (CCA) was performed using R with the community ecology package "vegan (2.0–4)" and combined with the adonis function of the R/vegan package (v2.6–2) using 10,000 permutations.

For the differential abundance analysis of the most prevalent taxa, taxa were aggregated at species, genus or phylum levels. Differential abundance was tested using ANOVA-like

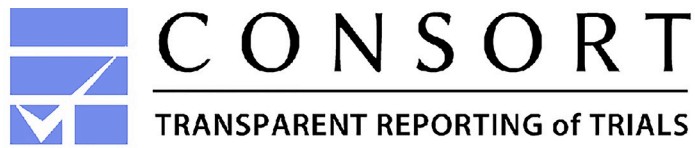

## CONSORT 2010 Flow Diagram

**Enrollment**

Assessed for eligibility (n= 33)

Excluded (n= 2)
♦ Not meeting inclusion criteria (n= 2)

Randomized (n= 31)

**Allocation**

Allocated to intervention (n= 17)
♦ Received allocated intervention (n= 17)
♦ Did not receive allocated intervention (n= 0)

Allocated to intervention (n= 14)
♦ Received allocated intervention (n= 14)
♦ Did not receive allocated intervention (n= 0)

**Follow-Up**

Discontinued intervention due to relapse of disease (n= 2), and COVID-19 mobility restrictions and perimeter lockdown (n= 3)

Discontinued intervention due to COVID-19 mobility restrictions and perimeter lockdown (n= 2)

**Analysis**

**Analyzed** (n= 12)
♦ Excluded from analysis (n= 0)

**Analyzed** (n= 11)
♦ Excluded from analysis due to unanalyzable samples (n= 1)

**Fig 1. Flow diagram.**

**Table 2. Baseline parameters by treatment group of recruited patients.**

| Characteristics | | Treatment group (n = 12) | Control group (n = 11) | *p*-value |
|---|---|---|---|---|
| Gender | | | | 0.855 |
| | Male | 5 (41.6%) | 5 (45.4%) | |
| | Female | 7 (58.3%) | 6 (54.5%) | |
| Age (years) | | 50.56 ± 11.65 | 45.40 ± 12.88 | 0.200 |
| Weight (Kg) | | 71.83 ± 16.35 | 73.82 ± 14.25 | 0.759 |
| Ethnic | | | | ND |
| | Caucasian | 11 (91.7%) | 11 (100%) | |
| | African | 1 (8.3%) | 0 (0.0%) | |
| Smoking | | | | 0.967 |
| | Yes | 1 (8.3%) | 1 (9.1%) | |
| | No | 6 (50.0%) | 6 (54.5%) | |
| | Former | 5 (41.6%) | 4 (36.4%) | |
| Partial mayo score | | | | 0.322 |
| | 0 | 3 (25.0%) | 2 (18.2%) | |
| | 1 | 5 (41.6%) | 2 (18.2%) | |
| | 2 | 4 (33.3%) | 7 (63.6%) | |
| IBS Rome IV type | | | | 0.838 |
| | Diarrhea | 1 (8.3%) | 2 (18.2%) | |
| | Mixed | 4 (33.3%) | 4 (33.3%) | |
| | Constipation | 3 (25.0%) | 2 (18.2%) | |
| | Unclassified | 4 (33.3%) | 3 (25.0%) | |
| Biologic treatment | | | | 0.662 |
| | Yes | 4 (33.3%) | 3 (27.3%) | |
| | No | 7 (58.3%) | 7 (63.6%) | |
| | NA | 1 (8.3%) | 1 (9.1%) | |
| Mesalazine treatment | | | | 0.580 |
| | Yes | 4 (33.3%) | 4 (9.1%) | |
| | No | 7 (58.3%) | 6 (54.5%) | |
| | NA | 1 (8.3%) | 1 (9.1%) | |
| Anxiety | | | | ND |
| | Normal | 12 (100%) | 11 (100%) | |
| | Border abnormal | 0 (0.0%) | 0 (0.0%) | |
| | Abnormal | 0 (0.0%) | 0 (0.0%) | |
| Depression | | | | 0.375 |
| | Normal | 9 (75.0%) | 9 (81.2%) | |
| | Border abnormal | 3 (25.0%) | 1 (9.1%) | |
| | Abnormal | 0 (0.0%) | 1 (9.1%) | |
| Adherence to Mediterranean diet | | 8.42 ± 2.39 | 7.36 ± 1.57 | 0.223 |

Parametric variables are expressed as mean ± SD for numerical data and as counts for categorical data. ND: not determined.

Differential Gene Expression Analysis (ALDEx2, v1.28.1, [60]). ALDEx2 performs centered log-ratio (CLR)-transformation to the count data for a compositionally coherent inference and estimates unadjusted and adjusted *p* values (controlling for Benjamini–Hochberg false-discovery rates (FDR)) from independent testing of Monte Carlo Dirichlet instances to control for type-I error due to the underestimated variance of low abundance taxa. We focused on results

**Table 3. Clinical questionnaires and parameters analyzed in CU patients suffering IBS symptoms with an 8-week bread dietary intervention (n = 23).**

| | Treatment group (n = 12) | | Control group (n = 11) | | Treatment *vs.* Control baseline (*p*-value) | Interaction effect | Main effects | |
|---|---|---|---|---|---|---|---|---|
| | Baseline w0 | EOS w8 | Baseline w0 | EOS w8 | | Intervention/Time (*p*-value) | Intervention (*p*-value) | Time (*p*-value) |
| **Partial Mayo** | | | | | 0.322 | 0.558 | 0.348 | 0.569 |
| 0 | 3 (25.0%) | 4 (33.3%) | 2 (18.2%) | 2 (18.2%) | | | | |
| 1 | 5 (41.6%) | 3 (25.0%) | 2 (18.2%) | 5 (45.5%) | | | | |
| 2 | 4 (33.3%) | 5 (41.6%) | 7 (63.6%) | 4 (36.4%) | | | | |
| **Abdominal pain** | | | | | 0.327 | 0.525 | 0.901 | <0.001* |
| Presence | 12 (100%) | 5 (41.6%) | 11 (100%) | 4 (36.4%) | | | | |
| Absence | 0 (0.0%) | 7 (58.3%) | 0 (0.0%) | 7 (63.6%) | | | | |
| **Likert scale** | | | | | 0.482§ | | 0.901 | |
| Worsening | | 0 (0.0%) | | 1 (9.1%) | | | | |
| Equal | | 5 (41.6%) | | 3 (27.3%) | | | | |
| Improvement | | 7 (58.3%) | | 7 (63.6%) | | | | |
| **IBS-SSS** | 210 ± 83.53 | 114.9 ± 85.26 | 239 ± 41.77 | 115.7 ± 78.87 | 0.456 | 0.400 | 0.547 | <0.001* |
| **Calprotectin (µg/g)** | 14.71 ± 10.36 | 23.60 ± 39.25 | 57.63 ± 72.94 | 51.83 ± 49.01 | 0.263 | 0.227 | 0.065 | 0.437 |
| **CRP (mg/dL)** | 0.31 ± 0.22 | 0.39 ± 0.31 | 0.19 ± 2.54 | 0.24 ± 0.20 | 0.624 | 0.435 | 0.719 | 0.925 |
| **Hemoglobin (g/dL)** | 14.14 ± 1.83 | 13.70 ±1.68 | 14.52 ± 1.03 | 14.64 ± 1.18 | 0.576 | 0.527 | 0.127 | 0.705 |
| **Albumin (g/dL)** | 4.53 ± 0.17 | 4.44 ± 0.21 | 4.42 ± 0.20 | 4.46 ± 0.14 | 0.103 | 0.195 | 0.368 | 0.689 |
| **Triglycerides (mg/dL)** | 155.55 ± 153.59 | 142.79 ± 87.51 | 109.85 ± 6.56 | 129.00 ± 137.18 | 1 | 0.643 | 0.377 | 0.933 |
| **Cholesterol (mg/dL)** | 195.33 ± 44.98 | 207.61 ± 44.45 | 196.85 ± 3.90 | 195.91 ± 36.87 | 0.902 | 0.595 | 0.666 | 0.615 |
| **ESR (mm)** | 15.45 ± 18.66 | 16.91 ± 16.49 | 11.60 ± 7.41 | 10.10 ± 5.74 | 0.524 | 0.724 | 0.198 | 0.991 |
| **Sodium (mEq/L)** | 139.62 ± 2.25 | 140.79 ± 2.61 | 140.63 ± 2.42 | 141.00 ± 1.00 | 0.455 | 0.521 | 0.366 | 0.208 |
| **Potassium (mEq/L)** | 4.79 ± 1.20 | 4.36 ± 0.29 | 4.75 ± 0.47 | 4.50 ± 0.31 | 0.423 | 0.599 | 0.906 | 0.145 |
| **Iron (mEq/L)** | 87.33 ± 49.92 | 80.78 ± 37.48 | 81.25 ± 21.06 | 92.60 ± 44.91 | 0.756 | 0.599 | 0.984 | 0.927 |
| **Chlorine (mEq/L)** | 102.43 ± 1.51 | 102.39 ± 1.84 | 102.33 ± 1.51 | 103.50 ± 1.78 | 0.882 | 0.455 | 0.738 | 0.350 |
| **Calcium (mEq/L)** | 9.42 ± 0.37 | 9.41 ± 0.49 | 9.53 ± 0.56 | 9.46 ± 0.29 | 1 | 0.846 | 0.421 | 0.768 |
| **Phosphorus (mEq/L)** | 3.34 ± 0.58 | 3.43 ± 0.51 | 3.39 ± 0.73 | 3.39 ± 0.62 | 0.867 | 0.735 | 0.427 | 0.860 |
| **Creatinine (mg/dL)** | 0.76 ± 0.16 | 0.77 ± 0.15 | 0.85 ± 0.14 | 0.85 ± 0.12 | 0.138 | 0.862 | 0.040* | 0.927 |

IBS-SSS: IBS Symptom Severity Score. CRP: C-Reactive Protein. ESR: Erythrocyte Sedimentation Rate. Parametric variables are expressed as mean ± SD for numerical data and as counts for categorical data. Linear mixed model (LMM) for numerical data and generalized linear mixed model (GLMM) were used for Intervention x Time interaction and main effects. * *p*-value ≤ 0.05.

from taxa with an overall prevalence of over 10%. Significant differences were considered when FDR-corrected p-value ≤ 0.05 (Welch's test).

In addition, the ratio between Firmicutes and Bacteroidetes (F/B ratio) phyla was calculated by dividing the relative abundances obtained by V3–V4 sequencing. Data normality was assessed through the Shapiro-Wilk test, and statistical differences were analyzed using the Wilcoxon rank-sum exact test or the Exact Wilcoxon Signed-Rank test.

### Ethics statement

The research has respected the fundamental principles of the Helsinki Declaration, meeting all the regulations of the Council of Europe and the Convention on Human Rights and Biomedicine, as well as the requirements established by the Spanish and Catalan legislation on biomedical research, data protection and bioethics. All participants have signed informed consent in compliance with Spanish data protection law (LO 3/2018, of December 5th of 2018, on Protection of personal data and digital rights guarantee (LOPDGDD) and published on December 6th 2018). The institutional review board and ethical committee approved the project on January 2019 (project code: RTC-2017-CU).

## Results

### Baseline characteristics

A total of 31 patients with quiescent UC who experienced moderate-to-severe IBS-like symptoms were recruited for the study. Eight patients withdrew from the study during the run-in period. Two of them discontinued the study due to worsening or relapse of UC, one patient was excluded due to unanalyzable samples, and the other five patients could not pick up the study bread due to COVID-19 mobility restrictions and perimeter lockdown in Spain from March to May of 2020 (Fig 1). The final 23 patients who completed the study had mean age of 47.84 (range 31–66 years), and among them, 13 patients (56%) were females, and 10 (44%) were males. Baseline characteristics by treatment group are shown in Table 2. Of note, none of the enrolled patients reported experiencing COVID-19 symptoms or testing positive for COVID-19 during the trial.

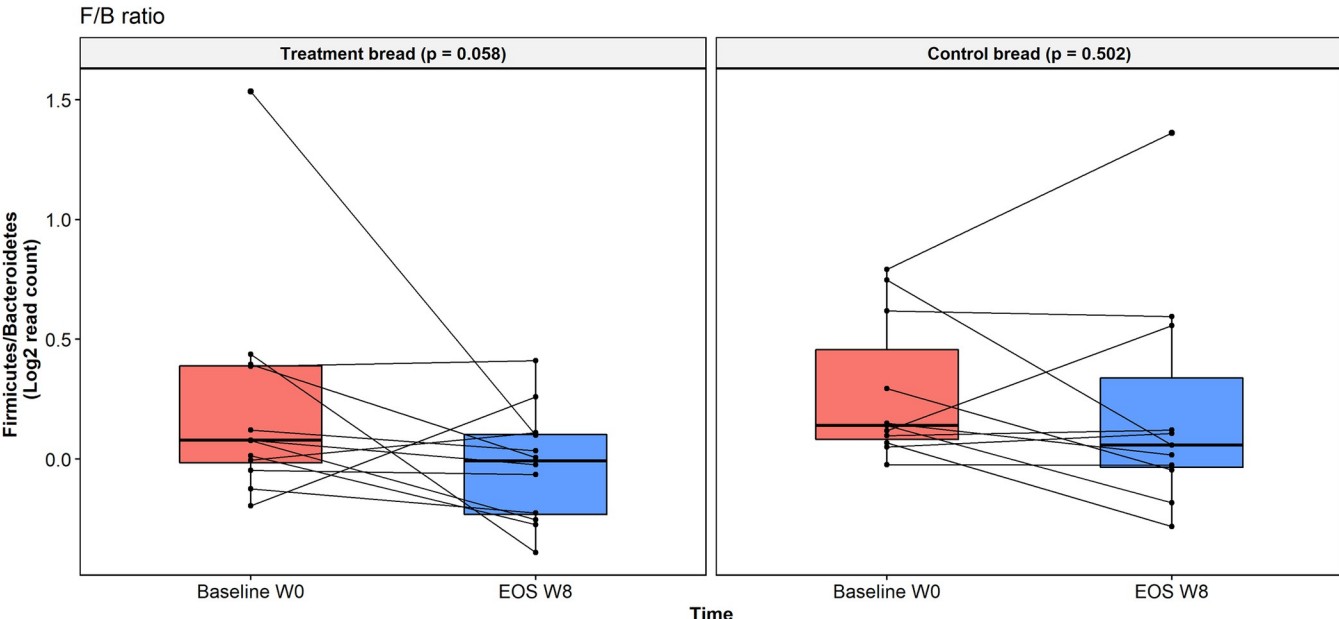

**Fig 2. Comparative of the Firmicutes/Bacteroidetes ratio according to groups of study.** Values are expressed as fold change (log2FC) between samples at baseline (Baseline w0) and samples at the end of the study (EOS w8) in the two groups of treatment. Boxes show median and interquartile range, and whiskers indicate the 5th to 95th percentile. Statistical differences were analyzed using Wilcoxon signed rank exact test.

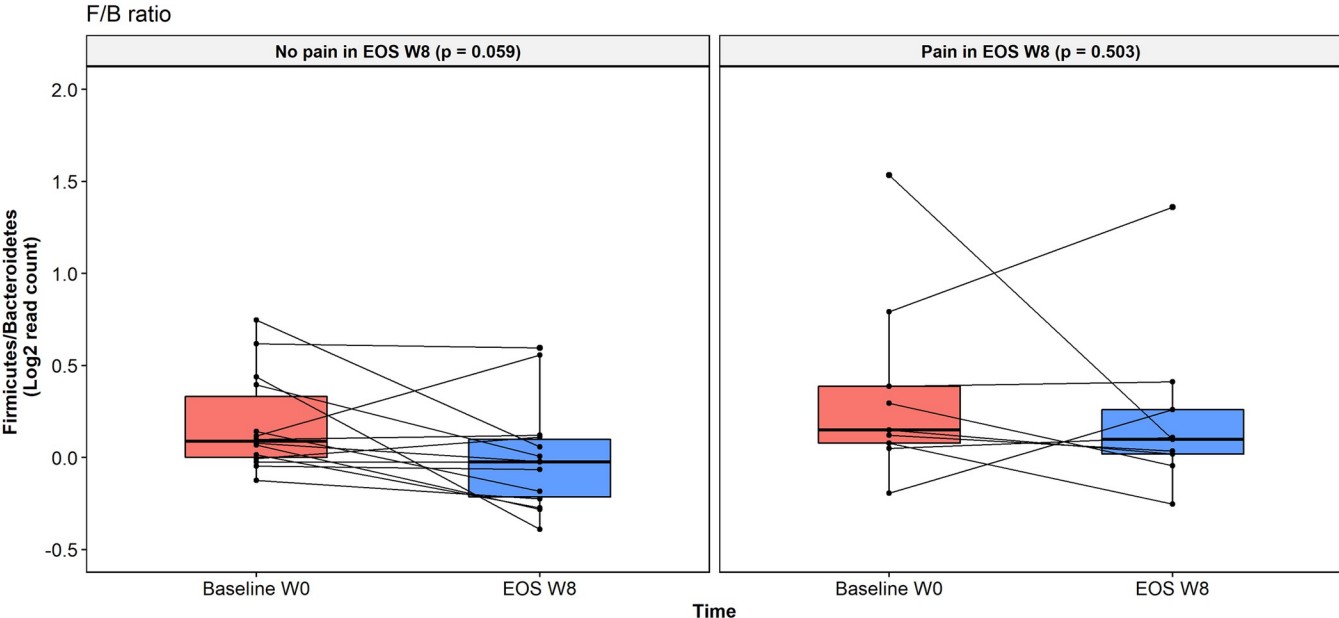

**Fig 3. Relationship of the Firmicutes/Bacteroidetes ratio with presence or absence of abdominal pain.** Values are expressed as fold change (log2FC). Boxes show median and interquartile range, and whiskers indicate 5th to 95th percentile. Statistical differences were analyzed using Wilcoxon signed rank exact test.

## Effect of bread diet on clinical symptomatology

Both linear mixed models and generalized linear mixed models were employed to examine the potential impact of the Intervention x Time interaction on various clinical variables (Table 3). However, our results did not reveal any statistically significant effects of this interaction across the variables assessed. Consequently, we proceeded to investigate the main effects, which unveiled noteworthy findings.

Specifically, we observed a significant time-effect on IBS-SSS scores ($p$-value < 0.001) and the presence of abdominal pain ($p$-value < 0.001). These changes were independent of the intervention. IBS-SSS scores exhibited a substantial decrease from 210.00 ± 83.53 to 114.90 ± 85.26 following treatment intervention and a similar reduction seen in the control group (from 239.00 ± 41.77 to 115.7 ± 78.87). It is to note that 9 out of 23 patients experienced a decrease of one grade in symptom severity (Severe > Moderate > Mild > Asymptomatic), with 6 patients from the treatment group and 3 from the control group. Additionally, 5 patients, 1 from the treatment group and 4 from the control group, reported a decrease of two grades, while 1 patient from the treatment group showed a remarkable improvement, moving from severe symptoms to an asymptomatic state, representing a decrease of 3 grades on the IBS-SSS scale.

Regarding the presence of abdominal pain, our findings indicated that 7 patients from the treatment bread group (58.33%) and 7 patients from the control bread group (63.63%) experienced complete relief of abdominal pain following the study intervention. However, we did not observe any other statistically significant differences for either the Time or Intervention effects (Table 3).

Notably, we evaluated patients' perceptions using a Likert scale, which revealed that 14 out of 23 patients reported either an improvement or a significant improvement in their symptomatology. This improvement was observed in 7 patients (58.33%) from the treatment group and 7 patients (63.63%) from the control group. Conversely, 8 patients did not notice any changes in their symptomatology, with 5 (41.66%) in the treatment bread group and 3 (27.27%) in the

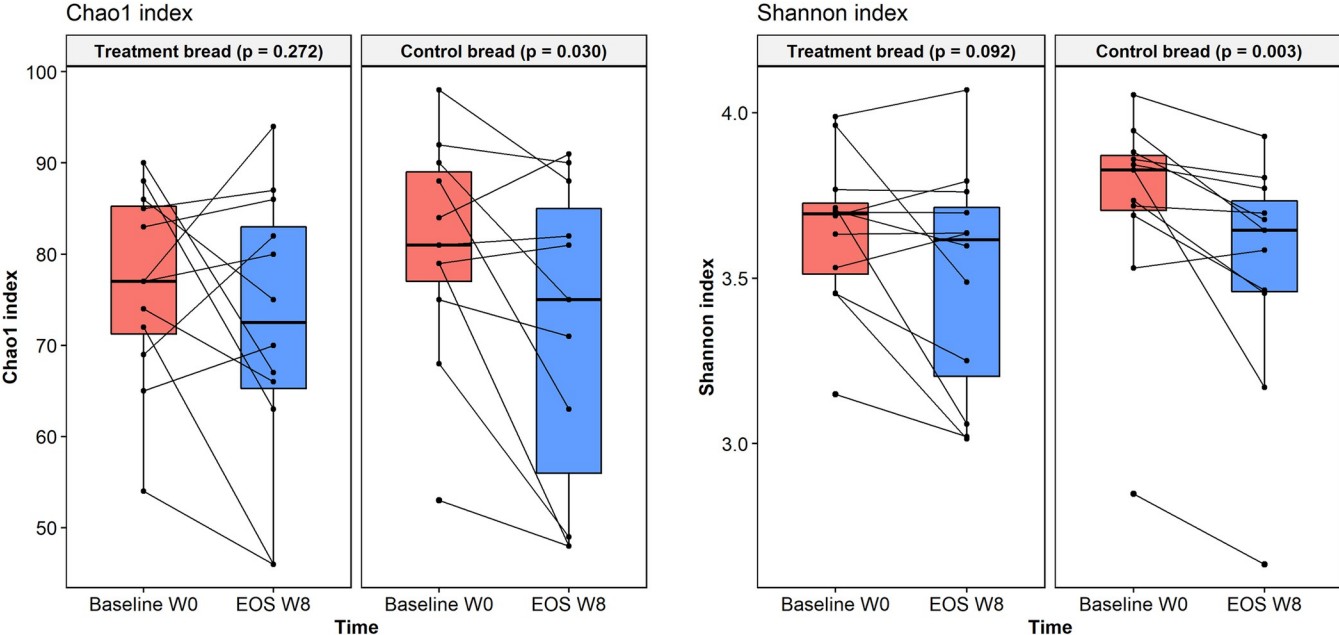

**Fig 4. Differences in alpha diversity analyses.** Chao1 and Shannon indices were compared between bread treatment group (n = 12) and control treatment group (n = 11). Boxes show median and interquartile range, and whiskers indicate 5th to 95th percentile. Statistical differences were analyzed using Paired t-test for the Chao1 index and Wilcoxon signed rank exact test for the Shannon index.

control bread group reporting no change. Only 1 patient from the control bread group (9.09%) reported worsened symptomatology (Table 3).

## Effect of bread diet on microbiota abundances and diversity

**Microbiota differential abundances.** To identify differences in bacterial abundances among groups, we used ALDEx2 at aggregated taxonomy levels (*i.e.*, phylum, genus, species). No statistically significant differential abundances were found in taxa with more than 10% of total prevalence at any taxonomic level (FDR-corrected *p-value* > 0.05; S1–S3 Tables).

The F/B ratio was not statistically different between groups at baseline ($p = 0.288$). The F/B ratio showed a suggestively decrease ($p = 0.058$) at the EOS visit compared to baseline in patients from the treatment bread group, but no changes in the control bread group were observed ($p = 0.502$) (Fig 2). We also observed that those patients without abdominal pain at the end of the trial, regardless of the study group, had a suggestive decrease in the ratio F/B ($p = 0.059$), but the ratio did not differ from baseline in patients with persistent abdominal pain ($p = 0.503$) (Fig 3).

The top 20 agglomerated phylum taxa were classified into 11 different phyla (S2 Fig), while the top 20 agglomerated genus taxa were classified into 20 different genera (S3 Fig). The most predominant phyla were Firmicutes and Bacteroidetes, while the most predominant genus was *Bacteroides*, followed by *Faecalibacterium* and *Blautia*.

## Microbiota diversity

Diversity indices presented high variability with null differences between pre-treatment groups (Chao1, $p = 0.422$; Shannon index, $p = 0.169$). Neither Chao1 nor Shannon indices presented differences in the treatment bread group when comparing samples at baseline versus EOS ($p = 0.272$ and $p = 0.092$, respectively) (Fig 4). Contrarily, both Chao1 ($p = 0.030$) and Shannon

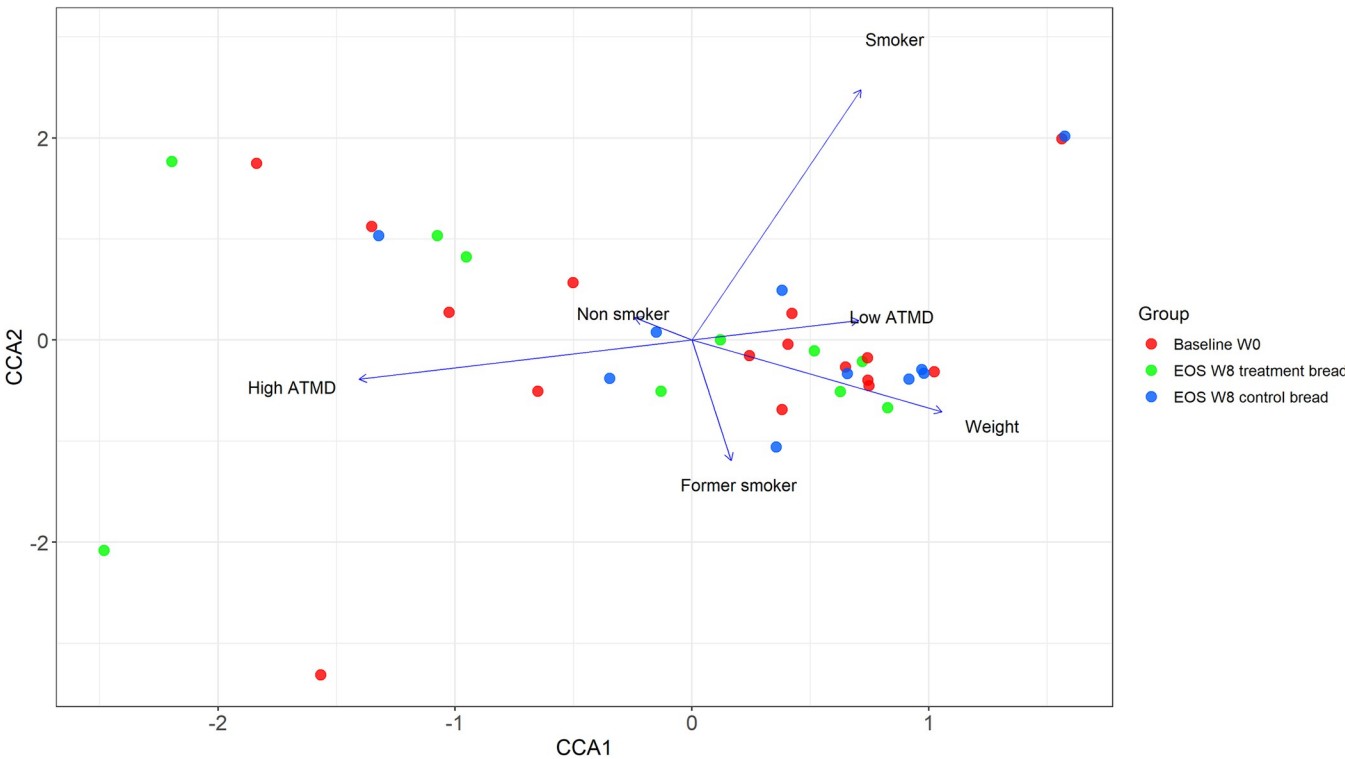

**Fig 5. Canonical correspondence analyses (CCA) biplot of the significant environmental variables.** Samples were colored by group of treatment (Baseline w0, samples at baseline week 0; EOS w8 treatment bread, samples at the end of study visit week 8 after treatment bread; and EOS W8 control bread, samples at the end of study visit week 8 after control bread). Blue arrows indicate an influence of significant variables on the plot. ATMD is the abbreviation of Adherence to Mediterranean diet. Eigenvalues: axis 1, 0.4782; axis 2, 0.4371.

indices ($p = 0.003$) were decreased at EOS compared to baseline in the control bread group, although the indices were not statistically lower than those from the treatment group (Chao1, $p = 0.956$; Shannon index, $p = 0.740$).

In all tested distance plots, samples did not cluster for group of treatment or sampling week (S4 Fig). No statistically significant values among pre and post-treatment groups were found in any distance metrics analyzed. Interestingly, the two principal coordinates in the weighted UniFrac distance (S4B Fig) explained 51.2% of the total variability of the data, indicating great dissimilarities between samples.

A canonical correspondence analysis (CCA) was conducted to further dissect the contributions of environmental variables to the microbial community structure (Fig 5). The variables "smoking", "weight", and "adherence to the Mediterranean diet (ATMD)" were identified to be the significant predictors across samples by the CCA model selection procedure ($F = 1.5472$, $p = 0.001$). According to the CCA profiles, the whole environmental parameters explained 0.570 (axis1, 0.298; axis 2, 0.272) of the variation in the species data. The analysis revealed a projection of the samples along the numerical variable "weight", although the subcategories of variables "smoking" and "ATMD" also segregated samples distribution on the plot.

## Discussion

### Primary clinical and microbiota outcomes

This study conducted a pilot clinical trial on twenty-three UC patients suffering from IBS-like symptoms to determine the impact of a traditional bread-based dietary intervention to

modulate intestinal dysbiosis and relieve symptoms. The key finding of this pilot study was that traditional bread intake did not greatly shift gut microbiome diversity, but slightly decreased the Firmicutes/Bacteroidetes ratio, which seemed to be associated with a relief of IBS-like symptoms. However, symptomatology relief was observed in both the treatment and control groups.

All participants remained in a clinical remission state during the trial. Therefore, the tested inflammatory parameters (fecal calprotectin, C-reactive protein and erythrocyte sedimentation rate) showed no significant differences compatible with disease relapse [61–63].

We observed a suggestive decrease in the F/B ratio among patients consuming the treatment bread and among patients that reported relief of abdominal pain at the end of the trial. While these changes did not reach statistical significance, they offer valuable insights into the potential impact and mechanisms through which dietary interventions, particularly those rich in fiber, can influence the gut microbiota of UC patients with IBS-like symptomatology. The F/B ratio has been reported to be increased in IBS patients compared to healthy subjects [64]. Other studies also observed an enrichment of Firmicutes abundance together with a reduced abundance of Bacteroidetes in IBS subjects compared to healthy individuals [64–66]. Members of the Bacteroidetes phylum are specialists in metabolizing dietary fibers, maximizing energy intake [67, 68]; while Firmicutes phylum contains some protease-producing bacteria [64]. This shift in the microbial composition might be associated with various aspects of IBS pathophysiology, including microinflammation of the colonic mucosa, increased levels of proteases and pro-inflammatory markers such as interleukin-6 (IL-6) and tumor necrosis factor-$\alpha$ (TNF-$\alpha$), and decreased levels of interleukin-10 (IL-10) [69].

Despite the limitations in statistical power, our study, in conjunction with others [70–73], supports the evidence that high-fiber diets increase the dietary fiber availability in the gut. This, in turn, may contribute restoring the F/B ratio imbalance and promoting the production of SCFA, to finally improve gut health and symptomatology.

## Microbiota differential abundances and diversity

Regarding diversity results, smoking, weight and adherence to the Mediterranean diet were the variables having a major impact on the microbiota diversity among samples. This observation is commonly found and supported by previous studies [74–76].

The most predominant bacterial phyla in patients from the present study were the Firmicutes and Bacteroidetes, while *Bacteroides*, *Faecalibacterium* and *Blautia* were the most abundant genera. The human gut microbiome is mainly constituted of bacteria belonging to Firmicutes, Bacteroidetes, Proteobacteria and Actinobacteria phyla [77]. At the genus level, other studies also reported *Bacteroides*, *Faecalibacterium* and *Blautia* as some of the most abundant in the human gut [78].

While prior research has suggested that dietary components like sourdough, long-fermentation, whole-grain flour, and sodium content [79, 80] could potentially influence gut microbiota [34, 36–39], our present study did not yield significant shifts in the gut microbiome. The differential abundance analyses conducted in the present study did not reveal any taxa showing significant differences after the dietary intervention at any taxonomic level analyzed. In both (*i.e.*, UC and IBS) clinical disorders, patients usually present an altered intestinal microbiota compared to healthy subjects [18–21], but data on patients with both coexisting diseases is scarce. Only two other studies have compared intestinal microbiota between quiescent UC patients with IBS-like symptoms and patients without symptoms [81, 82]. In agreement with our results, they did not observe any significant difference in the relative abundance of single bacterial taxa between these groups of patients using high-throughput sequencing.

## Limitations and strengths

The major strength of the present study was a double-blind, randomized controlled trial basis, which means that investigators, caregivers and patient were unaware of treatment assignment. Among the limitations of this study, we must highlight the choice of our control treatment, which was crafted from refined wheat flour, with a low proportion of sourdough starter, and shortly fermented for 2 hours. Even though the bread making technique of this bread was very different from that of the testing bread, they were nevertheless quite similar in terms of final nutritional composition. In a previous *in vitro* study, we observed that a similar bread baked using modern elaboration also increased the production of SCFA by the fecal commensal microbiota present from IBD patients [37]. This observation could explain the improvement in IBS symptomatology in the control group. The SCFA have been suggested as regulators of gut permeability [83], the inflammatory response [84], and as key mediators in microbiota-gut-brain interactions [85]. Therefore, they can have an essential role in the presence of the psychological factors (*e.g.*, increase in anxiety, depression, and reduced quality of life, among others) associated with IBS-like symptoms [2, 6–8]. Another limitation of the study is the reduced sample size, which might prevent the study from showing more robust results. Finally, no additional dietary instructions were provided during the trial, which could be affecting the observed results. Furthermore, although an 8-weeks timeframe might be a good time intervention, studies with longer duration will favour the identification of major microbiome shifts [86, 87].

While the generalizability of our study may be influenced by the unique study population, dietary intervention, and challenges posed by the COVID-19 pandemic, we rigorously followed standard procedures for data collection and analysis. Our pilot study findings lay a valuable foundation for future investigations with improved statistical power. Notably, our findings in UC patients with IBS symptoms may extend to those with IBS without co-existing UC, broadening the potential applicability of our results.

Overall, the present study provides a promising basis for dietary interventions in UC patients with IBS-like symptoms. The observed reduction in the F/B ratio and symptom improvement in both the traditional and control groups suggest the potential benefits of personalized dietary approaches. While immediate clinical recommendations may not be derived from our findings, they underscore the importance of tailoring dietary guidance for these patients. Future large-scale trials are needed to validate these results, but they point to a potential avenue for clinicians and healthcare providers to explore bread-based dietary interventions as a complementary strategy for managing IBS-like symptoms in UC patients.

## Conclusions

This is the first study to describe and compare the effect of a traditional bread-based dietary intervention on the gut microbiota composition and the relief of IBS-like symptoms in UC patients in clinical remission. Our findings suggest that consumption of traditional bread is associated with a decrease in the Firmicutes/Bacteroidetes ratio, potentially related to alleviating IBS-like symptoms. These findings, while promising, underscore the need for larger-scale research, a setting with clearly distinct breads in terms of composition, higher control of the participants' usual diet, and a control group of patients will be needed to better understand the potential benefits of different bread-making processes as dietary interventions to help improve gut health in quiescent UC with IBS-like symptoms.

## Supporting information

**S1 Checklist. CONSORT checklist.**
(PDF)

**S1 Table. Differential abundance analysis of the taxa with an overall prevalence over 10% aggregated at phylum taxonomic level.** Est: Estimate; SD: Standard deviation; P: p-value; Padj: FDR-corrected p-value.
(XLSX)

**S2 Table. Differential abundance analysis of the taxa with an overall prevalence over 10% aggregated at genus taxonomic level.** Est: Estimate; SD: Standard deviation; P: *p*-value; Padj: FDR-corrected *p*-value.
(XLSX)

**S3 Table. Differential abundance analysis of the taxa with an overall prevalence over 10% aggregated at species taxonomic level.** Est: Estimate; SD: Standard deviation; P: *p*-value; Padj: FDR-corrected *p*-value.
(XLSX)

**S1 Fig. Study design.** Intervention arms, baseline and end of study (EOS) visits, and reported outcome measures assessed at each time point. [†]IBS-SSS, IBS-Symptom Severity Score, [‡]HADS, Hospital Anxiety and Depression Scale, [§]ATMD, Adherence to Mediterranean Diet questionnaire.
(TIF)

**S2 Fig. Relative abundance plot for the top 20 most retrieved ASVs aggregated at phylum level.** Samples were classified by group of treatment (Baseline w0, samples at baseline week 0; EOS w8 treatment bread, samples at the end of study visit week 8 after treatment bread; and EOS w8 control bread, samples at the end of study visit week 8 after control bread).
(TIF)

**S3 Fig. Relative abundance plot for the top 20 most retrieved ASVs aggregated at genus level.** Samples were classified by group of treatment (Baseline w0, samples at baseline week 0; EOS w8 treatment bread, samples at the end of study visit week eight after treatment bread; and EOS w8 control bread, samples at the end of study visit week 8 after control bread).
(TIF)

**S4 Fig. Beta diversity analyses.** Principal coordinate analysis (PCoA) and Principal component analysis (PCA) with PERMANOVA tests of gut microbiota from the stool samples clustered in groups (Baseline w0, samples at baseline week 0; EOS w8 treatment bread, samples at the end of study visit week 8 after treatment bread; and EOS w8 control bread, samples at the end of study visit week 8 after control bread). Represented distances are based on unweighted (A) and weighted (B) UniFrac, Bray-Curtis dissimilarities (C), and Aitchison distances (D).
(TIF)

**S1 Protocol. Trial protocol.**
(PDF)

## Acknowledgments

The authors would like to thank all study participants for contributing to this research. The authors would also like to thank all researchers of the inflammatory digestive diseases and microbiota group for their contribution to the present work.

## Author Contributions

**Conceptualization:** Josep Oriol Miquel-Cusachs, Sílvia Delgado-Aros, Librado Jesús Garcia-Gil, Isidre Elias, Xavier Aldeguer.

**Data curation:** Aleix Lluansí.

**Formal analysis:** Aleix Lluansí.

**Funding acquisition:** Sílvia Delgado-Aros, Isidre Elias, Xavier Aldeguer.

**Investigation:** Aleix Lluansí, Marc Llirós, Robert Carreras-Torres.

**Methodology:** Aleix Lluansí, Marc Llirós, Librado Jesús Garcia-Gil.

**Project administration:** Sílvia Delgado-Aros, Librado Jesús Garcia-Gil, Isidre Elias, Xavier Aldeguer.

**Resources:** Aleix Lluansí, Anna Bahí, Montserrat Capdevila, Anna Feliu, Laura Vilà-Quintana, Núria Elias-Masiques, Emilio Cueva, Laia Peries, Leyanira Torrealba, Josep Oriol Miquel-Cusachs, Míriam Sàbat, David Busquets, Carmen López, Isidre Elias, Xavier Aldeguer.

**Supervision:** Marc Llirós, Robert Carreras-Torres, Librado Jesús Garcia-Gil, Isidre Elias, Xavier Aldeguer.

**Visualization:** Xavier Aldeguer.

**Writing – original draft:** Aleix Lluansí.

**Writing – review & editing:** Marc Llirós, Robert Carreras-Torres, Sílvia Delgado-Aros, Librado Jesús Garcia-Gil, Isidre Elias, Xavier Aldeguer.

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
