## [Decision Letter · Decision Letter 0]

4 Oct 2023

PONE-D-23-21156Impact of bread diet on intestinal dysbiosis and irritable bowel syndrome symptoms in quiescent ulcerative colitis: A pilot studyPLOS ONE

Dear Dr. Lluansí,

Thank you for submitting your manuscript to PLOS ONE. After careful consideration, we feel that it has merit but does not fully meet PLOS ONE’s publication criteria as it currently stands. Therefore, we invite you to submit a revised version of the manuscript that addresses the points raised during the review process.

We look forward to receiving your revised manuscript.

Kind regards,

Miquel Vall-llosera Camps

Staff Editor

PLOS ONE

Journal Requirements:

6. Please ensure that you refer to Figure 5 in your text as, if accepted, production will need this reference to link the reader to the figure.

**Additional Editor Comments: **

I would like to sincerely apologise for the delay you have incurred with your submission. It has been exceptionally difficult to secure reviewers to evaluate your study. We have now received three completed reviews; the comments are available below. The reviewers have raised significant scientific concerns about the study that need to be addressed in a revision.

Please revise the manuscript to address all the reviewer's comments in a point-by-point response in order to ensure it is meeting the journal's publication criteria. Please note that the revised manuscript will need to undergo further review, we thus cannot at this point anticipate the outcome of the evaluation process.

Reviewers' comments:

Reviewer's Responses to Questions

**Comments to the Author**

1. Is the manuscript technically sound, and do the data support the conclusions?

Reviewer #1: Partly

Reviewer #2: Yes

Reviewer #3: Partly

2. Has the statistical analysis been performed appropriately and rigorously? 

Reviewer #1: No

Reviewer #2: Yes

Reviewer #3: No

3. Have the authors made all data underlying the findings in their manuscript fully available?

Reviewer #1: Yes

Reviewer #2: Yes

Reviewer #3: Yes

4. Is the manuscript presented in an intelligible fashion and written in standard English?

Reviewer #1: Yes

Reviewer #2: Yes

Reviewer #3: Yes

5. Review Comments to the Author

Reviewer #1: Within the manuscript under review Lluansí et al present the results concerning the pilot study in patients with ulcerative colitis at the symptomless phase, where they investigate the impact of traditionally baked bread compared with the bread baked with modern baking methods. The concept of the current study is to investigate if the bread with long fermentation could induce prebiotic capacity to relief of the symptoms and improve the gastrointestinal comfort within the population studied. The potential of the breads are chosen based on the results of in vitro GI model modification of the gut microbiota. Thus, also this in vivo study is hypothetised to introduced changes in the gastrointestinal microbiota composition closer to healthy population. Although the concept and the study is representing an important concept and is timely, there are still major concerns that need to taken into account.

Of note: English language check will be needed.

Design and methods:

It is important that the patients with UC in the remission were studied. However, the description of the design is unclear. The authors write the activities in ‘both visits’, but there is no indication before that if the design is parallel or cross-over design. This should be very clear in the beginning of the explanation of the design. It is suggested that the description is moved as the first chapter in the material and methods -section to allow reader to better follow the flow of the study and decisions made in the recruitment. In addition, it is important that the figure of the design is part of the main document, not as supplement. In dietary interventions the design and justification underlying the choice of the design are elemental to enable clear evaluation of the results.

It is of note that the study may have been improved remarkably if the group of healthy population would have been studied as well. Already in healthy population there is a large variation in the intestinal microbiota composition, and the variability increases along the disease impact, especially in the populations studied here. Thus, it is not possible to really state that the changes would have been caused by the actual bread consumption.

The description of the drop outs should also be included in the methods section. It is a pity that the study ended up with overall number of 23 participants, that does not allow - in practical terms – many conclusion with such a challenging population studied here.

Regarding the bread, it is of interest why the sodium content is clearly higher than in the target bread? There are indications that high salt intake might have an impact on the intestinal microbiota, and thus, might also have an impact here, especially with low number of the participants.

Statistical procedure of the clinical and biochemical measures is quite vague. It is understood that with such a low number of participants one, in principal, should use non-parametric methods or be careful with the normal distribution of the variables. However, the long term study should be analysed fikrst for the time x group -interactions and time point or within group differences. Thus, the analyses such as mixed model time trends should be used to see if there are any differences between the groups. With the R-package used here it is fairly simple to do and to present for the reader.

Results:

Baseline characteristics in the whole 31 population and treatment groups based characteristics in the group of 23 participants need to be shown in the main document, not in the supplement. Why are these information that is essential for the reader hidden in the supplement?

It is reported that in both treatment groups the same number (7) of participants had complete relief of the abdominal pain. What might have caused this impact? It does not support the hypothesis and thus, needs to be carefully discussed.

Give explanations for all the abbreviations as table’s footnotes – some of them are missing.

Discussion:

The authors state at the beginning of the discussion: ‘The key finding of this pilot study was that traditional bread intake decreased the Firmicutes/Bacteroidetes ratio, which seemed to be associated with a relief of IBS-like symptoms. However, symptomatology relief was observed in both the treatment and control groups.” Was this really the finding? The statistical significance is missing, although near the significance and there was no difference when compared to the control bread group. In addition the relief of symptoms was happening in both groups that is clearly stated. The statement is not convincing based on the present results. In addition, the results related with the diversity have to be summarized also in the beginning of the Discussion. It is appreciated that later in the Discussion it is clearly written that other factors explain the differences found in the diversity measures.

Limitations:

This kind of dietary study cannot really be double blinded – breads will differ either in their appearance and or their taste. So this statement needs to be discarded.

As the study is introduced as the pilot study, how the results seem for the authors? Do these results indicate the start of the main study? This could be clearly discussed and justified in addition to the usual text in the conclusion regarding the need of further studies.

Reviewer #2: Dear Authors,

I have critically appraised your paper titled "Impact of Bread Diet on Intestinal Dysbiosis and Irritable Bowel Syndrome Symptoms in Quiescent Ulcerative Colitis: A Pilot Study." While your study explores an exciting topic, several areas would benefit from revision and clarification.

1. Baseline Difference in Diets:

Please provide more detailed information regarding the baseline dietary habits of your study participants, as differences in diets can influence gut microbiota and may confound your results.

2. Clinical vs. Endoscopic vs. Deep Remission:

Consider discussing the differences and implications of clinical, endoscopic, and deep remission, as these distinct states may have varying effects on gut microbiota and symptomatology. Were all recruited patients only in clinical remission, or did some have endoscopic and histological remission as well, and how does this impact the findings?

3. Impact of Lockdown and COVID-19:

Since the study was conducted during lockdown and the COVID-19 pandemic, please address whether any participants had COVID-19 during the trial, as COVID-19 and its treatment could potentially impact gut microbiota and symptoms.

4. Lack of Detailed Bread Composition:

Provide a comprehensive analysis of the bread's composition, including fiber content and prebiotic components, to elucidate the dietary factors influencing the gut microbiota.

5. Short Duration:

Eight weeks may be too short of a follow-up period and might not reflect the sustainability of the microbiota changes, as short follow-up may be one of the limitations of this study.

6. Discussion of Non-significant Findings:

Provide a more in-depth discussion of non-significant findings, explaining their potential biological relevance and considering the study's statistical power.

7. Clinical Implications:

Expand on the clinical implications of your research by discussing how the findings may impact the management of UC patients with IBS-like symptoms in a practical clinical setting.

8. Gender Imbalance:

Address the gender imbalance (F>>M) in your study and discuss its potential implications on the results

9. Calprotectin Values:

Discuss the clinical significance of the increased calprotectin values observed after the treatment bread intervention and its potential implications

10. Generalizability:

Address the limitations in the generalizability of your findings, particularly how they may apply to a broader range of patient populations and dietary patterns.

11. Symptom Relief in Control Group:

Clarify the implications of symptom relief observed in control group, as this may impact the specificity of the traditional bread intervention.

I believe that addressing these points will significantly enhance the quality and clarity of your study. Please consider these suggestions for revision in your manuscript. I look forward to reviewing the revised version of your paper.

Reviewer #3: Major Revision:

Table 2: Test the interaction of time points by intervention group rather than repeatedly applying t-tests. If the interaction effect is significant, provide an interpretation of the results. Do not test main effects because the tests for main effects are uninteresting in light of significant interactions. If interaction effects are non-significant, drop the interaction effects from the model and test the main effects. Determining which results to present when testing interactions is often a multi-step process.

Minor Revisions:

1- Define s.d. at its first appearance. Typically standard deviation is abbreviated SD.

2- Indicate the date range subjects were enrolled in the study.

3- Line 210: Clarify if Pearson’s tests were used to compare categorical data between the intervention groups.

4- State and justify the study’s target sample size with a pre-study statistical power calculation. The power calculation should include: (1) the estimated outcomes in each group; (2) the α (type I) error level; (3) the statistical power (or the β (type II) error level); (4) the target sample size and (5) for continuous outcomes, the standard deviation of the measurements.

5- Thoroughly proofread the document.

6. PLOS authors have the option to publish the peer review history of their article (what does this mean?). If published, this will include your full peer review and any attached files.

Reviewer #1: No

Reviewer #2: No

Reviewer #3: No

---

## [Author Response · Author response to Decision Letter 0]

14 Nov 2023

Response to the editor/reviewers- “Impact of bread diet on intestinal dysbiosis and irritable bowel syndrome symptoms in quiescent ulcerative colitis: A pilot study” (PONE-D-23-21156)

We thank the editor for their positive response and consideration of our manuscript for publication. We fully addressed all the points raised by the academic editor and reviewers on a point-by-point basis. Please, find below our responses:

Academic editor:

1- “Please ensure that your manuscript meets PLOS ONE's style requirements, including those for file naming”

Response: We have meticulously reviewed and made necessary adjustments to ensure that our manuscript complies with PLOS ONE's style requirements.

2- “We note that the grant information you provided in the ‘Funding Information’ and ‘Financial Disclosure’ sections do not match.”

Response: We have carefully rechecked the grant information provided in the 'Funding Information' section within the manuscript, and we believe that the information is identical to that of the “Financial Disclosure”. However, we have identified an error in the submission process that led to the discrepancy in the ministry's name. The correct information is the one provided within the manuscript. The funds were received from the “Ministerio de Economía y Competitividad”, which was known as the “Ministerio de Economía, Industria y Competitividad” at the time of receiving the grant. 

3- “In your Data Availability statement, you have not specified where the minimal data set underlying the results described in your manuscript can be found. PLOS defines a study's minimal data set as the underlying data used to reach the conclusions drawn in the manuscript and any additional data required to replicate the reported study findings in their entirety. All PLOS journals require that the minimal data set be made fully available. For more information about our data policy, please see http://journals.plos.org/plosone/s/data-availability.

We will update your Data Availability statement to reflect the information you provide in your cover letter.”

Response: Minimal data is provided as main manuscript and supplementary information tables.

4- “We note that you have stated that you will provide repository information for your data at acceptance. Should your manuscript be accepted for publication, we will hold it until you provide the relevant accession numbers or DOIs necessary to access your data. If you wish to make changes to your Data Availability statement, please describe these changes in your cover letter and we will update your Data Availability statement to reflect the information you provide.”

Response: We have uploaded repository information, including all raw reads generated in the present study and associated metadata, however we have not made them available until publication. We apologize for inconveniences to both editorial staff and reviewers. Data will be accessible under accession number: PRJNA902141. We include a private URL for reviewers to gain access to the data: 

https://dataview.ncbi.nlm.nih.gov/object/PRJNA902141?reviewer=3pocuvj91987qcs9b5jd3e4m3c

5- “Your ethics statement should only appear in the Methods section of your manuscript. If your ethics statement is written in any section besides the Methods, please move it to the Methods section and delete it from any other section. Please ensure that your ethics statement is included in your manuscript, as the ethics statement entered into the online submission form will not be published alongside your manuscript.”

Response: We have relocated the ethics statement to the Methods section of our manuscript, as per editor’s guidance.

6- “Please ensure that you refer to Figure 5 in your text as, if accepted, production will need this reference to link the reader to the figure.”

Response: We have rectified the typo concerning the reference to Figures 4 and 5 in the revised manuscript. The text now correctly refers to Figures 4 and 5, as intended.

7- “Please include captions for your Supporting Information files at the end of your manuscript, and update any in-text citations to match accordingly. Please see our Supporting Information guidelines for more information: http://journals.plos.org/plosone/s/supporting-information.”

Response: We have ensured that all captions in the main manuscript and supplementary information are located at the end of the manuscript, as per your guidelines, and updated the in-text citations to match accordingly.

Reviewer 1:

1- “Of note: English language check will be needed.”

Response: We appreciate the reviewer's feedback and have carefully reviewed the entire manuscript for language improvements. We believe the changes made have enhanced the overall clarity and readability of the manuscript. 

2- “Design and methods:

It is important that the patients with UC in the remission were studied. However, the description of the design is unclear. The authors write the activities in ‘both visits’, but there is no indication before that if the design is parallel or cross-over design. This should be very clear in the beginning of the explanation of the design. It is suggested that the description is moved as the first chapter in the material and methods -section to allow reader to better follow the flow of the study and decisions made in the recruitment. In addition, it is important that the figure of the design is part of the main document, not as supplement. In dietary interventions the design and justification underlying the choice of the design are elemental to enable clear evaluation of the results.

It is of note that the study may have been improved remarkably if the group of healthy population would have been studied as well. Already in healthy population there is a large variation in the intestinal microbiota composition, and the variability increases along the disease impact, especially in the populations studied here. Thus, it is not possible to really state that the changes would have been caused by the actual bread consumption.

The description of the drop outs should also be included in the methods section. It is a pity that the study ended up with overall number of 23 participants, that does not allow - in practical terms – many conclusion with such a challenging population studied here.

Regarding the bread, it is of interest why the sodium content is clearly higher than in the target bread? There are indications that high salt intake might have an impact on the intestinal microbiota, and thus, might also have an impact here, especially with low number of the participants.

Statistical procedure of the clinical and biochemical measures is quite vague. It is understood that with such a low number of participants one, in principal, should use non-parametric methods or be careful with the normal distribution of the variables. However, the long term study should be analysed fikrst for the time x group -interactions and time point or within group differences. Thus, the analyses such as mixed model time trends should be used to see if there are any differences between the groups. With the R-package used here it is fairly simple to do and to present for the reader.”

Response: We appreciate reviewer 1 comments with respect to design and methods of our study. We have taken the following actions:

• We have clarified the study design at the beginning of the sub-section “Study design” (lines 111-114). 

• We have relocated the entire sub-section “Study design” to the beginning of the Methods section. We believe this adjustment will allow readers to better follow the flow of the study. 

• A detailed description of the study design was included as supplementary information (S1 Fig), providing a comprehensive explanation of our approach. Regarding the figure placement, we would like to clarify that in accordance with PLOS ONE's style requirements, we designated 'Figure 1' for the flow diagram, which includes enrollment schedule, allocation and follow-up. This flow diagram is essential for depicting the study's progression and adherence to protocols. In addition, considering the overall content within the main text, we have already included five figures and three tables. Given PLOS ONE's formatting guidelines, we would like to ensure that our submission aligns with their requirements while maintaining the clarity and comprehensiveness of our study presentation. 

• We have refined the clarity of our study's aim, which is to examine the potential effects of bread consumption on the symptoms of individuals with coexisting IBS and UC (lines 114-116). We have explicitly highlighted that our study design intentionally focused on this particular patient population, thereby not considering the inclusion of a control group of subjects.

• We included a description of drop-outs in the “Baseline characteristic” sub-section (lines 318-322) of the Methods, as well as a schematic representation in Figure 1 (flow diagram). 

We also share the concern about finalizing the study with a total of 23 participants. While we recognize the challenges of working with a limited sample size in this complex population, we have conducted the study to the best of our ability within the constraints of the available participants and acknowledged the limitations of the sample size and its impact on the discussion and conclusions in the manuscript.

• We have included some words in the Discussion section (lines 473-476) regarding the potential effect of high sodium intake on the intestinal microbiota and the non-significant shifts observed in the gut microbiome in our study.

• We have conducted linear mixed model and generalized mixed model analyses specifically examining the interaction between intervention and time and controlling for individual variability. We have incorporated the results of these analyses into our revised manuscript, specifically in sub-section “Effect of bread diet on clinical symptomatology” of Results and in Table 3 (formerly Table 2 in the previous version of the manuscript). We have also added the statistical approach used in the Methods section (lines 231-239). We believe that these revisions will substantially strengthen the statistical foundation of our study and provide a more comprehensive understanding of the data. 

3- “Results:

Baseline characteristics in the whole 31 population and treatment groups based characteristics in the group of 23 participants need to be shown in the main document, not in the supplement. Why are these information that is essential for the reader hidden in the supplement?

It is reported that in both treatment groups the same number (7) of participants had complete relief of the abdominal pain. What might have caused this impact? It does not support the hypothesis and thus, needs to be carefully discussed.

Give explanations for all the abbreviations as table’s footnotes – some of them are missing.”

Response: To better clarify the findings in the study, we have taken following actions:

• We have included baseline characteristics information as Table 2 in the sub-section “Baseline characteristics” in the Results (line 328) to make it more accessible for the reader. In addition, we have added more information in this table regarding other concomitant clinical treatments (i.e., biologic and mesalazine treatments). 

• In our discussion, we considered several potential explanations to the unexpected relief of abdominal pain in both intervention groups (lines 488-495). These factors encompass the similarity in nutritional parameters of both breads despite significant differences in their dough preparation methods, constraints imposed by our relatively small sample size, and the potential influence of uncontrolled environmental variables such as diet, smoking, and weight. In addition, in a previous in vitro study, we observed that the bread baked using modern elaboration proxies (here used as control) also increased the production of short-chain fatty acids by the fecal commensal microbiota present in IBD patients (see reference 38 from the manuscript). We firmly believe that these limitations discussed in the manuscript offer valuable insights into this particular observation.

• We provided detailed description of unexplained abbreviations as footnotes in Table 3 (former table 2 in the unrevised manuscript)

4- “Discussion:

The authors state at the beginning of the discussion: ‘The key finding of this pilot study was that traditional bread intake decreased the Firmicutes/Bacteroidetes ratio, which seemed to be associated with a relief of IBS-like symptoms. However, symptomatology relief was observed in both the treatment and control groups.” Was this really the finding? The statistical significance is missing, although near the significance and there was no difference when compared to the control bread group. In addition the relief of symptoms was happening in both groups that is clearly stated. The statement is not convincing based on the present results. In addition, the results related with the diversity have to be summarized also in the beginning of the Discussion. It is appreciated that later in the Discussion it is clearly written that other factors explain the differences found in the diversity measures.”

Response: We appreciate Reviewer 1 insights. We have clarified the key finding statement in the Discussion section to better align with the results (lines 436-437). We hope these changes enhance the clarity and accuracy of our manuscript.

5- “Limitations:

This kind of dietary study cannot really be double blinded – breads will differ either in their appearance and or their taste. So this statement needs to be discarded.

As the study is introduced as the pilot study, how the results seem for the authors? Do these results indicate the start of the main study? This could be clearly discussed and justified in addition to the usual text in the conclusion regarding the need of further studies.”

Response: We took specific measures to minimize these differences, ensuring that both types of bread had similar appearance and taste when supplied to participants. By doing so, our intention was to prevent both participants and investigators/caregivers from distinguishing between the two types of bread based on taste or appearance, thus avoiding a predisposition to change by study subjects. We strongly believe that the double-blinding is one of the strengths of the present study. 

We revised our conclusions and made some adjustments to highlight the preliminary nature of this study and the need for subsequent investigations (lines 515-522).

Reviewer 2:

1- “Baseline Difference in Diets:

Please provide more detailed information regarding the baseline dietary habits of your study participants, as differences in diets can influence gut microbiota and may confound your results.”

Response: Thank you for your comments and for revising the manuscript. While we agree that differences in diets can influence gut microbiota and may potentially confound our results, we focused on assessing the impact of the specific bread-based dietary intervention on the modification of UC with IBS-like symptomatology. As such, we did not collect extensive data on participants' dietary habits beyond the study's dietary intervention and adherence to Mediterranean diet. However, participants were asked to not alter their diet during the intervention period (lines 120-121).

2- “Clinical vs. Endoscopic vs. Deep Remission:

Consider discussing the differences and implications of clinical, endoscopic, and deep remission, as these distinct states may have varying effects on gut microbiota and symptomatology. Were all recruited patients only in clinical remission, or did some have endoscopic and histological remission as well, and how does this impact the findings?”

Response: In this study, our primary objective was to investigate the effects of a traditional bread-based dietary intervention on the gut microbiota and symptomatology in UC patients in clinical and endoscopic remission, defined as a total Mayo score of lower than 2. We have clarified this along the new manuscript. By focusing on this specific remission state, we aimed to create a targeted and homogeneous cohort that would facilitate recruitment and study design. However, we acknowledge the potential variations in microbiota and symptomatology associated with other remission states, such as histological or deep remission. This study lays the foundation for future research that can delve into the distinctions between these remission states to gain a more comprehensive understanding of their effects. 

3- “Impact of Lockdown and COVID-19:

Since the study was conducted during lockdown and the COVID-19 pandemic, please address whether any participants had COVID-19 during the trial, as COVID-19 and its treatment could potentially impact gut microbiota and symptoms.”

Response: We confirm that none of the participants reported experiencing COVID-19 symptomatology or testing positive for COVID-19 during the trial. This aspect was not explicitly stated in the initial version of the manuscript, and we have now added a statement in the methodology section to clarify this (line 329). 

4- “Lack of Detailed Bread Composition:

Provide a comprehensive analysis of the bread's composition, including fiber content and prebiotic components, to elucidate the dietary factors influencing the gut microbiota.”

Response: We did collect data on the nutritional composition, including fiber content, for both types of bread, and this information is provided in Table 1. However, we acknowledge that a more detailed analysis of prebiotic components would have been beneficial, and we plan to consider this in future research. We appreciate the feedback and will ensure that future investigations provide a more in-depth assessment of dietary factors. Moreover, we have identified an error in the description of the components involved in modern bread production. We have subsequently refined this definition to specify that modern bread incorporates a minimal proportion of sourdough starter (line 203). We would like to emphasize that this change is purely a clarification of terminology and does not impact the core content, findings, or discussion presented in the manuscript.

5- “Short Duration:

Eight weeks may be too short of a follow-up period and might not reflect the sustainability of the microbiota changes, as short follow-up may be one of the limitations of this study.”

Response: We have expanded our discussion (lines 498-499) to note that longer-duration studies could be favorable for the identification of more substantial microbiome changes. We selected an 8-week duration based on findings from previous studies (e.g., Dong TS et al., 2020; Nutrients), expecting it to be sufficient to observe microbiota changes in response to the dietary intervention. However, longer diet durations may be necessary to identify major microbiome shifts.

6- “Discussion of Non-significant Findings:

Provide a more in-depth discussion of non-significant findings, explaining their potential biological relevance and considering the study's statistical power.”

Response: We have expanded the sub-section “Primary clinical and microbiota outcomes” of the Discussion (Lines 446-449 and 455-460) to better explain the potential relevance of our non-significant findings.

7- “Clinical Implications:

Expand on the clinical implications of your research by discussing how the findings may impact the management of UC patients with IBS-like symptoms in a practical clinical setting.”

Response: We have expanded our discussion (lines 512-520) regarding the clinical implications of our research to better understanding its practical significance.

8- “Gender Imbalance:

Address the gender imbalance (F>>M) in your study and discuss its potential implications on the results”

Response: We have detected and corrected a typo in lines 323 and 324 regarding the percentage of males and females in the study. In our S1 Table, we reported that there were no significant differences in gender within both study groups (5/7 vs. 5/6). As such, we considered that the potential implications of gender imbalances on the results would not apply in this particular study, and the findings can be interpreted without gender-related bias.

9- “Calprotectin Values:

Discuss the clinical significance of the increased calprotectin values observed after the treatment bread intervention and its potential implications”

Response: Upon revisiting the clinical parameters with a different statistical approach, as suggested by Reviewers 1 and 3, it was discovered that the previously observed increase in calprotectin values is no longer considered significant (Table 3). Consequently, we have incorporated these refined results into the revised manuscript. 

10- “Generalizability:

Address the limitations in the generalizability of your findings, particularly how they may apply to a broader range of patient populations and dietary patterns.”

Response: We have included a statement in the limitations of our manuscript (lines 506-511) to address the limitations in the generalizability. We believe that the findings from our pilot study provide valuable insights and lay the foundation for future investigations with greater statistical power.

11- “Symptom Relief in Control Group:

Clarify the implications of symptom relief observed in control group, as this may impact the specificity of the traditional bread intervention.”

Response: In our discussion, we considered the implication of symptom relief observed in control group and highlighted several potential explanations. These factors encompass the similarity in nutritional parameters of both breads despite significant differences in their dough preparation methods, constraints imposed by our relatively small sample size, and the potential influence of uncontrolled environmental variables such as diet, smoking, and weight. In addition, in a previous study, we observed in vitro that the bread baked using modern elaboration (here used as control) also increased the production of SCFA by the fecal commensal microbiota present in IBD patients (see reference 38 from the manuscript). These limitations discussed in the manuscript (lines 485-494) offer valuable insights into this particular observation. However, we agree that future research should consider incorporating additional control measures to differentiate the specific effects of the traditional bread intervention from other potential influencing factors.

Reviewer 3:

1- “Major Revision:

Table 2: Test the interaction of time points by intervention group rather than repeatedly applying t-tests. If the interaction effect is significant, provide an interpretation of the results. Do not test main effects because the tests for main effects are uninteresting in light of significant interactions. If interaction effects are non-significant, drop the interaction effects from the model and test the main effects. Determining which results to present when testing interactions is often a multi-step process.”

Response: We would like to extend our gratitude for the Reviewer 3 valuable comments and for revising the manuscript. 

• We have conducted linear mixed model and generalized mixed model analyses specifically examining the interaction between intervention and time and controlling for individual variability. Since all interaction effects were non-significant, we refined the model by excluding the interaction term, focusing solely on main effects. 

• We have incorporated the results of these analyses into our revised manuscript, particularly sub-section “Effect of bread diet on clinical symptomatology” in Results and in Table 3 (formerly Table 2 in the previous manuscript). 

• We have also added the statistical approach used in the methods section (lines 223-244 of the revised manuscript). We believe that these revisions will substantially strengthen the statistical foundation of our study and provide a more comprehensive understanding of the data. 

2- “Minor Revisions:

1- Define s.d. at its first appearance. Typically standard deviation is abbreviated SD.”

Response: We have replaced “s.d.” by "SD" throughout the revised manuscript. Additionally, we have provided a clear definition of "SD" at its first appearance in the text, precisely at line 215.

2- “Indicate the date range subjects were enrolled in the study.”

Response: We have specified the date range of participation at line 164 of the revised manuscript.

3- “Line 210: Clarify if Pearson’s tests were used to compare categorical data between the intervention groups.”

Response: We have clarified that Pearson’s chi-squared test was used to compare categorical data (line 231 of the revised manuscript).

4- “State and justify the study’s target sample size with a pre-study statistical power calculation. The power calculation should include: (1) the estimated outcomes in each group; (2) the α (type I) error level; (3) the statistical power (or the β (type II) error level); (4) the target sample size and (5) for continuous outcomes, the standard deviation of the measurements.”

Response: We have provided additional details regarding the sample size calculation at lines 218-226. It's worth emphasizing that to the best of our knowledge, this is the first study on a dietary interventions involving bread among UC individuals suffering IBS-like symptomatology. Consequently, the availability of directly comparable data from previous studies is limited. In this study, we formulated our sample size calculation by drawing insights from previous research, although we encountered difficulties in reaching the initially estimated sample size. Therefore, we wish to emphasize that the results of this study should be considered preliminary in nature, and we classify it as a pilot study. It is important to keep this classification in mind when evaluating the results.

5- “Thoroughly proofread the document.”

Response: We have thoroughly proofread the revised manuscript.

---

## [Decision Letter · Decision Letter 1]

14 Dec 2023

PONE-D-23-21156R1Impact of bread diet on intestinal dysbiosis and irritable bowel syndrome symptoms in quiescent ulcerative colitis: A pilot studyPLOS ONE

Dear Dr. Lluansí,

Thank you for submitting your manuscript to PLOS ONE. After careful consideration, we feel that it has merit but does not fully meet PLOS ONE’s publication criteria as it currently stands. Therefore, we invite you to submit a revised version of the manuscript that addresses the points raised during the review process.

We look forward to receiving your revised manuscript.

Kind regards,

Chia-Yen Dai

Academic Editor

PLOS ONE

Journal Requirements:

Reviewers' comments:

Reviewer's Responses to Questions

**Comments to the Author**

1. If the authors have adequately addressed your comments raised in a previous round of review and you feel that this manuscript is now acceptable for publication, you may indicate that here to bypass the “Comments to the Author” section, enter your conflict of interest statement in the “Confidential to Editor” section, and submit your "Accept" recommendation.

Reviewer #1: All comments have been addressed

Reviewer #2: All comments have been addressed

Reviewer #3: (No Response)

2. Is the manuscript technically sound, and do the data support the conclusions?

Reviewer #1: Yes

Reviewer #2: Yes

Reviewer #3: Yes

3. Has the statistical analysis been performed appropriately and rigorously? 

Reviewer #1: Yes

Reviewer #2: Yes

Reviewer #3: Yes

4. Have the authors made all data underlying the findings in their manuscript fully available?

Reviewer #1: Yes

Reviewer #2: Yes

Reviewer #3: Yes

5. Is the manuscript presented in an intelligible fashion and written in standard English?

Reviewer #1: Yes

Reviewer #2: Yes

Reviewer #3: Yes

6. Review Comments to the Author

Reviewer #1: The modifications made based on the review as well as questions made have been answered with great effort - this is highly appreciated. It is clearly visible that the manuscript has greatly improved and its scientific quality and reliability as well as clarity for the readers have markedly been increased.

Reviewer #2: (No Response)

Reviewer #3: Minor revisions:

1- Line 119: Grammatical correction: Furthermore, subjects were asked not to alter their diet during the intervention period.

2- Line 163: Remove "during December 2019 and August 2021". The sentence that follows provides more clarity.

3- Table 1: Identify the statistical testing method(s) used to estimate the p-values in Table 1. To improve clarity, consider moving Table 1 after the "Statistical analysis of blood test and questionnaire data" section. This section seems to include the statistical testing methods used to estimate p-values in Table 1.

4- Line 220: Indicate the statistical testing method which achieves 80% power. Perhaps it is the t-test.

5- Line 231: Indicate the underlying covariance structure used in the Linear Mixed Models and the criteria for selecting it.

6- Tables 2 and 3: A) In addition to the frequencies, provide percentages that correspond to them. B) State the sample sizes of the groups in the header row.

7. PLOS authors have the option to publish the peer review history of their article (what does this mean?). If published, this will include your full peer review and any attached files.

Reviewer #1: No

Reviewer #2: No

Reviewer #3: No

---

## [Author Response · Author response to Decision Letter 1]

20 Dec 2023

Response to the editor/reviewers- “Impact of bread diet on intestinal dysbiosis and irritable bowel syndrome symptoms in quiescent ulcerative colitis: A pilot study” (PONE-D-23-21156)

We thank the editor for their positive response and consideration of our manuscript for publication. We fully addressed all the points raised by the academic editor and reviewers on a point-by-point basis. Please, find below our responses:

Academic editor:

“Please review your reference list to ensure that it is complete and correct. If you have cited papers that have been retracted, please include the rationale for doing so in the manuscript text, or remove these references and replace them with relevant current references. Any changes to the reference list should be mentioned in the rebuttal letter that accompanies your revised manuscript. If you need to cite a retracted article, indicate the article’s retracted status in the References list and also include a citation and full reference for the retraction notice”

Response: We have carefully reviewed the reference list and ensured that it is complete and correct.

Reviewer 1:

“The modifications made based on the review as well as questions made have been answered with great effort - this is highly appreciated. It is clearly visible that the manuscript has greatly improved and its scientific quality and reliability as well as clarity for the readers have markedly been increased.”

Response: We appreciate the Reviewer 1 comments and agree with the improvement of the scientific quality of our manuscript. 

Reviewer 3:

Minor revisions:

1- “Line 119: Grammatical correction: Furthermore, subjects were asked not to alter their diet during the intervention period.”

Response: We thank Reviewer 3 for their valuable comments and for revising the manuscript. We have corrected the grammatical error as suggested.

2- “Line 163: Remove "during December 2019 and August 2021". The sentence that follows provides more clarity.”

Response: We have removed the sentence as suggested.

3- “Table 1: Identify the statistical testing method(s) used to estimate the p-values in Table 1. To improve clarity, consider moving Table 1 after the "Statistical analysis of blood test and questionnaire data" section. This section seems to include the statistical testing methods used to estimate p-values in Table 1.”

Response: We have specified the statistical test used in Table 1 to estimate the p-values (lines 213-215). While we acknowledge the potential enhancement in clarity by relocating Table 1 after the statistical analysis section, we have positioned it immediately after its first mention following requirements stipulated by the journal.

4- “Line 220: Indicate the statistical testing method which achieves 80% power. Perhaps it is the t-test.”

Response: We have included the statistical test used in the revised manuscript (line 221 of the revised manuscript).

5- “Line 231: Indicate the underlying covariance structure used in the Linear Mixed Models and the criteria for selecting it.”

Response: In our Linear Mixed Models (LMM) for analyzing the effects the intervention, we incorporated a random intercept term with a variance components structure to capture subject-specific variability. This choice facilitated the modeling of subject-specific variability while maintaining a straightforward and interpretable model. Model fit diagnostics supported the appropriateness of the selected covariance structure. We have specified the covariance structure used in the manuscript (line 235).

6- “Tables 2 and 3: A) In addition to the frequencies, provide percentages that correspond to them. B) State the sample sizes of the groups in the header row.”

Response: We have included the percentages and sample sizes in Tables 2 and 3 as suggested.

---

## [Decision Letter · Decision Letter 2]

9 Jan 2024

PONE-D-23-21156R2Impact of bread diet on intestinal dysbiosis and irritable bowel syndrome symptoms in quiescent ulcerative colitis: A pilot studyPLOS ONE

Dear Dr. Lluansí,

Thank you for submitting your manuscript to PLOS ONE. After careful consideration, we feel that it has merit but does not fully meet PLOS ONE’s publication criteria as it currently stands. Therefore, we invite you to submit a revised version of the manuscript that addresses the points raised during the review process.

We look forward to receiving your revised manuscript.

Kind regards,

Chia-Yen Dai

Academic Editor

PLOS ONE

Journal Requirements:

Reviewers' comments:

Reviewer's Responses to Questions

**Comments to the Author**

1. If the authors have adequately addressed your comments raised in a previous round of review and you feel that this manuscript is now acceptable for publication, you may indicate that here to bypass the “Comments to the Author” section, enter your conflict of interest statement in the “Confidential to Editor” section, and submit your "Accept" recommendation.

Reviewer #3: (No Response)

2. Is the manuscript technically sound, and do the data support the conclusions?

Reviewer #3: Yes

3. Has the statistical analysis been performed appropriately and rigorously? 

Reviewer #3: Yes

4. Have the authors made all data underlying the findings in their manuscript fully available?

Reviewer #3: Yes

5. Is the manuscript presented in an intelligible fashion and written in standard English?

Reviewer #3: Yes

6. Review Comments to the Author

Reviewer #3: Minor revisions:

1- Table 1: If data is normally distributed, summarize using means and standard deviations. If data is non-normally distributed, summarize using medians, first and third quartiles.

2- Tables 2 & 3: Consider displaying each level for categorical factors in separate rows rather than by separating with backslashes. This is the typical style for displaying categorical data in tabular form.

7. PLOS authors have the option to publish the peer review history of their article (what does this mean?). If published, this will include your full peer review and any attached files.

Reviewer #3: No

---

## [Author Response · Author response to Decision Letter 2]

10 Jan 2024

Minor revisions:

1- Table 1: If data is normally distributed, summarize using means and standard deviations. If data is non-normally distributed, summarize using medians, first and third quartiles.

Response: We thank Reviewer 3 for their valuable comments and for revising the manuscript. We have modified Table 1 as suggested.

2- Tables 2 & 3: Consider displaying each level for categorical factors in separate rows rather than by separating with backslashes. This is the typical style for displaying categorical data in tabular form.

Response: We appreciate Reviewer’s valuable comment regarding Table 2 and 3 style. We have modified categorical factors to display each level as a separate row as suggested. Furthermore, table footnote has also been modified accordingly.

---

## [Editor Report · Decision Letter 3]

14 Jan 2024

Impact of bread diet on intestinal dysbiosis and irritable bowel syndrome symptoms in quiescent ulcerative colitis: A pilot study

PONE-D-23-21156R3

Dear Dr. Lluansí,

We’re pleased to inform you that your manuscript has been judged scientifically suitable for publication and will be formally accepted for publication once it meets all outstanding technical requirements.

Kind regards,

Chia-Yen Dai

Academic Editor

PLOS ONE

Additional Editor Comments (optional):

No further queries
---

## [Editor Report · Acceptance letter]

9 Feb 2024

PONE-D-23-21156R3 

PLOS ONE

Dear Dr. Lluansí, 

I'm pleased to inform you that your manuscript has been deemed suitable for publication in PLOS ONE. Congratulations! Your manuscript is now being handed over to our production team.

Kind regards, 

on behalf of

Dr. Chia-Yen Dai 

Academic Editor

PLOS ONE